# Ramie Fabric Treated with Carboxymethylcellulose and Laser Engraved for Strain and Humidity Sensing

**DOI:** 10.3390/mi13081309

**Published:** 2022-08-13

**Authors:** Shangxuan Shi, Jiao Liang, Chenkai Qu, Shangbi Chen, Bin Sheng

**Affiliations:** 1School of Optical Electrical and Computer Engineering, University of Shanghai for Science and Technology, Shanghai 200093, China; 2Shanghai Key Laboratory of Modern Optical Systems, Engineering Research Center of Optical Instruments and Systems, Shanghai 200093, China; 3Shanghai Aerospace Control Technology Institute, Shanghai 200233, China

**Keywords:** ramie fabric, laser engraving, CMC, strain sensor, humidity sensor

## Abstract

Wearable fabric sensors have attracted enormous attention due to their huge potential in human health and activity monitoring, human–machine interaction and the Internet of Things (IoT). Among natural fabrics, bast fabric has the advantage of high strength, good resilience and excellent permeability. Laser engraving, as a high throughput, patternable and mask-free method, was demonstrated to fabricate fabric sensors. In this work, we developed a simplified, cost-effective and environmentally friendly method for engraving ramie fabric (a kind of bast fabric) directly by laser under an ambient atmosphere to prepare strain and humidity sensors. We used carboxymethylcellulose (CMC) to pretreat ramie fabric before laser engraving and gained laser-carbonized ramie fabrics (LCRF) with high conductivity (65 Ω sq^−1^) and good permeability. The strain and humidity sensors had high sensitivity and good flexibility, which can be used for human health and activity monitoring.

## 1. Introduction

Electronic textiles (E-textiles) have been widely used in recent years owing to their functionality in sensing [1,2,3,4,5], energy harvesting [1,6,7,8] and wireless transmission [9,10,11]. In terms of efficient signal detecting, the fabric sensor plays a significant role and attracts enormous attention due to its ubiquitous nature of fabrics such as low cost, high toughness, good wearability, non-invasive manner of sensing and excellent permeability [12]. Nowadays, the fabric sensor is applied in various fields, such as human health and activity monitoring [13,14,15,16], human–machine interaction [17,18] and the Internet of Things (IoT) [19,20].

To date, the major technology of functionalizing fabric to fabricate fabric sensors includes screen printing [21,22], inkjet printing [23,24], physical vapor deposition (PVD) [25], pyrolysis [26,27], electrophoretic deposition [28], and others [29]. Inkjet printing has several advantages including mask-free fabrication, high print resolution, and scalability from table-top devices to big press units. The screen-printing technique has the capability to print electronic devices at a low cost, with very little or no material wastage [30]. Conductive metals (e.g., Cu, Au, Pt, Ag) can be deposited on fabrics by PVD, making it possible to produce flexible and lightweight electroconductive textiles. Pyrolysis is a simple and effective method to carbonize the fabric in an oven and prepare highly flexible strain sensors [31,32]. Electrophoretic deposition is performed from an aqueous dispersion at ambient temperatures under a direct current (DC) electric field, which is rapid, repeatable, and environmentally benign. However, these aforementioned methods for preparing fabric sensors still have certain limitations. In inkjet printing, it takes time-consuming steps to obtain ink with appropriate viscosity and surface tension to prevent nozzle blockage. The screen-printing process provides high wet film thickness, resulting in high spreading of ink and low resolution if not cured instantly. PVD can only be carried out in a high vacuum environment and pyrolysis needs the conditions of high temperature and a protective atmosphere. Electrophoretic deposition takes time to disperse and the formation of gases due to electrolysis creating bubbles, leading to non-uniform deposition on the fabrics. As a high throughput, patternable, and mask-free method, laser engraving was demonstrated to fabricate carbonized fabric for preparing fabric sensors [33,34,35,36,37]. Some chemically synthesized fabrics, such as PI [35] and Kevlar fabric [34], can be engraved directly by laser to synthesize graphene under an ambient atmosphere, without any pretreatment.

However, when some natural materials (cotton, bast, wood and silk) were engraved by laser under an ambient atmosphere, they are ablated and become fragile due to violent oxidation, resulting in poor conductivity and inapplicability to the preparation of flexible sensors. In order to solve the above problem, many methods [36,38,39,40] had been developed in different ways. One direct method is to use a controlled atmosphere chamber to allow Ar/H_2_ to flow through the chamber in order to isolate oxygen and avoid natural materials experiencing ablation [38]. Another strategy of multiple lasing and defocusing the laser has also proved to be an effective method [39]. Moreover, using fire retardant to treat samples seems to be a simpler and more economical method [36,39]. Although there are the aforementioned effective solutions, these laser engraving methods on natural materials still have certain limitations. The controlled atmosphere chamber required for isolating oxygen increases the cost and multiple lasing and defocusing of the laser involves complicated preparation routes. The addition of commercial flame retardants containing phosphorus or boron may harm the environment to a certain extent. As a result, exploring a low-cost, simplified and environmentally friendly method to engrave natural fabrics by laser and fabricate a fabric sensor is necessary.

Among natural fabrics, bast fabric, which is made from the fiber of natural bast plants (rich in cellulose and lignin), mainly includes jute, ramie, flax, and hemp fabrics, which have the advantage of high strength, good resilience and excellent permeability [41]. Previously, Liang heated linen fabric with graphene oxide at 900 °C under the protection of nitrogen and then integrated the fabric with silver nanowires to prepare a wearable strain sensor [42]. Liu carbonized the hemp fabrics to 800 °C with a nitrogen atmosphere and fabricated the permeable pressure sensors [43]. In Liu’s other work, a high-performance stretchable strain sensor based on linen fabric was developed through a similar carbonization method and polymer-assisted copper deposition [44]. However, the carbonization of bast fabric can only be carried out under the conditions of high temperature and a protective atmosphere, and can not be patterned and functionalized for some sensing applications. As a result, it is a very attractive challenge to engrave bast fabric by laser under an ambient atmosphere for preparing the bast fabric sensor.

In this work, a simplified, cost-effective and environmentally friendly method was developed for engraving patterns on ramie fabric (a kind of bast fabric) directly under ambient atmosphere to fabricate strain and humidity sensors based on laser-carbonized ramie fabrics (LCRF). We used carboxymethylcellulose (CMC) to pretreat ramie fabric before laser engraving and gained LCRF with high conductivity (65 Ω sq^−1^) and good flexibility. We carried out a series of comparative tests and proved that CMC pretreatment has a good effect on the flame retardancy of fabrics during laser engraving. In order to demonstrate the ability of this method in fabricating flexible sensors on ramie fabric, wearable strain sensors and humidity sensors based on LCRF were fabricated. Compared with the traditional high-temperature carbonizing fabric in the oven, this method has the advantages of low equipment requirements, no protective gas atmosphere, low cost, less time-consuming, and customization.

## 2. Materials and Methods

### 2.1. Materials

Ramie fabric (100% ramie, ≈700 μm of thickness) was made from the stem fiber of natural ramie plant, which was purchased from Huidian company (Guangzhou, China). CMC powder (which contains 9.9 wt% of sodium) was used to treat ramie fabric, which was purchased from Sinopharma Chemical Reagent Co., Ltd. (Shanghai, China). Silver conductive glue used to connect copper foil with LCRF was purchased from Shenzhen Ausbond Co., Ltd. (Guangdong, China). NaCl, LiCl_2_, MgCl, NaBr, KCl and K_2_SO_4_ for preparation of saturated solution were purchased from Sinopharma Chemical Reagent Co., Ltd. (Shanghai, China).

### 2.2. LCRF Fabrication by Laser Engraving

LCRF was fabricated following the workflow shown in Figure 1a. We first fully soaked a piece of ramie fabric (≈4 × 4 cm^2^, Figure 1b) in CMC solution (range from 0.5 to 3.9 wt% in deionized water) for one hour and dried it in a vacuum oven at 80 °C for two hours. After drying, we directly engraved ramie fabric via irradiation using a semiconductor laser (450 nm wavelength, 0.30 mm spot width, from DAJA, Dongguan, China) under ambient atmosphere. The laser beam is a Gaussian beam, and the area far from the center of the laser receives less energy in each scan. The width of the overlapping area in consecutive passes is about 0.14 mm, which ensures that the area far from the center of the laser can also obtain enough energy in consecutive passes. The surface resistance value of carbonized fabric in this paper was measured through a four-point probe meter (HPS2526, Changzhou, China). In each surface resistance measurement, we chose the average surface resistance of five samples as the surface resistance value under the same condition. The error distributions of the samples tested in each case ranged from 2% to 4%. Figure 1c,d show that the laser-engraved area turned black and the generated LCRF has high integrity and conductivity (surface resistance is around 10^2^ Ω sq^−1^). We also observed untreated ramie fabric engraved by laser and the generated LCRF had low-degree carbonation and poor conductivity (surface resistance is around 10^6^ Ω sq^−1^, Figure 1e,f).

Then, we weighed the ramie fabric before and after soaking to obtain the mass change and tested the effect of adding CMC with different mass percentages on the conductivity of LCRF under fixed laser scan speed and power (37% maximum laser power of 15 W and scan speed = 36 mm s^−1^, Figure 1g). When the mass percentage of CMC on ramie fabric initially increased, the conductivity of LCRF significantly improved. When the mass percentage of CMC is greater than 4.48%, the surface resistance can be stably maintained at the level of 10^2^ Ω sq^−1^. We found that when the mass percentage of CMC was 5.62% (gained from 2.1 wt% CMC solution), the surface resistance had a minimum value (105 Ω sq^−1^), which was about 10^4^ times lower than carbonized untreated ramie fabric. Considering the operability of the performance tests, all further characterization and experiments were carried out when the ramie fabric was soaked in 2.1 wt% CMC solution for one hour.

### 2.3. Laser Operation Parameter Optimization for LCRF Fabrication

The LCRF with the lowest surface resistance can be obtained by adjusting the laser power percentage and laser scan speed. Laser power is varied from 31% to 41% of maximum operational value (4.65–6.15 W, respectively), with the laser scan speed varying from 12–48 mm s^−1^. Figure 2a,b show the optical photographs of samples (1 × 1 cm^2^) representing different combinations of parameters and their surface resistance values, respectively. At 33% low power, the carbonization of fabric is insufficient and there are areas of the fabric that are not carbonized (Figure 2(c1)). High laser power of up to 40% ensures that there is enough energy for the conversion of cellulose and lignin fibers into LCRF with good conductivity (surface resistance ≈ 75 Ω sq^−1^, Figure 2(c2)), which can also be achieved at low scan speed. Moreover, there appears to be a limit above which the energy imparted on the fabric begins to damage its fibers significantly (Figure 2(c3)). When the scanning speed is as low as 12 mm s^−1^ or 24 mm s^−1^, increasing the laser power is more likely to lead to ablation. The samples will be ablated at 37% or 38% power, and the lowest surface resistance is about 241 Ω sq^−1^. When the scanning speed increases to 36 mm s^−1^ or 48 mm s^−1^, the samples will be ablated at 40% power. As a result, appropriately increasing the scanning speed will help to obtain more samples at higher power and reduce the surface resistance of the samples. The lowest surface resistance of all samples (≈65 Ω sq^−1^) is obtained when the power = 39% and scan speed = 36 mm s^−1^.

### 2.4. Laser Engraving of Strain and Humidity Sensor on Ramie Fabric

We firstly used laser to engrave a rectangular pattern (2 × 6 cm^2^) and an interdigital pattern (2 × 2 cm^2^) on the ramie fabric and obtained the LCRF. The patterning of LCRF for strain and humidity sensors was controlled by a translational platform (voidmicro, DAJA). The laser power percentage and scanning speed for the laser engraving were 39% and 36 mm s^−^^1^, respectively. After laser engraving, the two ends of rectangular patterned LCRF were connected with copper foil and silver glue. Then, the sample was dried at 80 °C for 2 h and we gained the strain sensor. We applied silver glue to bottom of the interdigital patterned LCRF, dried it at 80 °C for 2 h as well and gained the humidity sensor. Finally, nine humidity sensors engraved on the ramie fabric could form a 3 × 3 humidity sensing array.

### 2.5. Performance Test of the Strain and Humidity Sensor

The resistance of the strain sensor was measured using a desktop digital multimeter (DMM6500, Tek Keithley Co., Ltd., Cleveland, OH, USA). The gauge factor (GF) of the sensor was calculated according to the equation: GF = (ΔR/R_0_)/ε(1)

In which ΔR, R_0_, and ε are the resistance variations, original resistance, and bending strains applied, respectively. According to the reported literature [45,46], the ε was determined based on the radius of curvature and thickness of the sensor under tension or compression. A high-precision single-axis electrodynamic force tester (ZQ-990B, Zhiqu Precision Instrument Co., Ltd., Dongguan, China) was used to bend and release the strain sensor repeatedly.

The capacitance of the humidity sensor was measured at room temperature (25 °C) using a desktop digital bridge (VC4092A, Xi’an Shengli Instrument Co., Ltd., Xi’an, China) with a frequency of 1 kHz, an AC voltage of 3.0 V and a recording interval of 1 s. According to the literature [47,48,49], six saturated salt solutions with specific relative humidity (RH), which were LiCl_2_ (11% RH), MgCl (32% RH), NaBr (57% RH), NaCl (75% RH), KCl (84% RH) and K_2_SO_4_ (97% RH), respectively were prepared as the performance test environment of the humidity sensor. The sensor’s sensitivity S is defined as follows:S = (C − C_0_)/∆RH(2)
where S is the sensitivity, C_0_ and C are the sensor’s capacitance in 11% RH and humidity environments, respectively, and ∆RH is the variation in RH. A high-precision single-axis electrodynamic force tester was used to bend and release the humidity sensor repeatedly (same as the strain sensor). The response and recovery times are defined as the time required to reach 90% of the change of sensor capacitance.

## 3. Results and Discussion

### 3.1. The Effect of CMC Treatment on Ramie Fabric

We further evaluated the effect of CMC treatment on ramie fabric under laser engraving. CMC, as a water-soluble derivative of cellulose, can be synthesized from some plant-based precursors (such as corn cobs, banana pseudo-stem and pineapple peel, etc.) and some waste materials (such as wastepaper, waste textiles and cotton gin wastes, etc.) [50]. Thermogravimetric analysis carried out in the air showed that after high-temperature treatment, the ramie fabric treated with CMC has a higher residual weight than that of untreated ramie fabric (Figure 3). The weight of the untreated ramie fabric was reduced to less than 1.0% by 460 °C. As a comparison, ramie fabric treated with CMC retained approximately 8.6% of its original weight even at 600 °C. The residual weight of the ramie fabric treated with CMC was considerably higher than the weight of Na^+^ (approximately 0.56 wt%) in the fabric, which might indicate the existence of carbonaceous materials. These results showed that the CMC treatment was conducive to avoiding ablation for ramie fabrics exposed to the air during laser engraving, which was consistent with the conclusion that the surface resistance of the ramie fabric significantly reduced more than 10^4^ times after the CMC treatment.

According to previous literature [51,52], Na^+^ can lower the activation energy of the dehydration step and help the growth of carbonaceous materials at high temperatures. The good effect on flame retardancy of CMC might be attributed to the presence of Na^+^ in CMC. In order to further research the effect of Na^+^ in CMC on fabric during the laser-engraving, a controlled experiment using NaCl to treat fabric was designed. We soaked the ramie fabric with NaCl solution and the mass percentage of Na^+^ on the ramie fabric was ~0.56 wt%, which was the same as the ramie fabric treated with CMC. Then, we engraved ramie fabric treated with NaCl in the same way as we did with the ramie fabric treated with CMC (power = 39% and scan speed = 36 mm s^−1^). The surface resistance of the carbonized ramie fabric treated with NaCl was ~4128 Ω sq^−1^, more than 240 times smaller than that of the carbonized untreated ramie fabric. Thermogravimetric analysis showed that the residual weight of ramie fabric treated with NaCl was approximately 6.3% at 600 °C, which was more than six times higher than that of the untreated ramie fabric. These results seemed to indicate that the addition of Na^+^ could avoid ablation and reduce the surface resistance of the carbonized ramie fabric. However, compared with the surface resistance of the carbonized ramie fabric treated with NaCl, the surface resistance of the carbonized ramie fabric treated with CMC was more than 63 times smaller, confirming that the CMC treatment had a better effect in reducing the surface resistance of the carbonized ramie fabric. Moreover, compared with NaCl, carbon and Na^+^ in CMC can synergistically play a better role in the flame retardant effect, which could explain why the residual weight of the ramie fabric treated with CMC is higher than the ramie fabric treated with NaCl. As a result, using CMC to pretreat ramie fabric is an effective method to avoid fabric ablation and gain LCRF with high conductivity.

### 3.2. LCRF Characterization

A thorough physicochemical characterization of LCRF was performed to retrieve information about its chemical and morphological properties. Taking the previous work of adjusting laser parameters as a reference, the LCRF fabricated by laser was selected (power = 39% and scan speed = mm s^−1^). The surface morphology of pristine ramie fabric and LCRF was observed by a scanning electron microscope (TESCAN MIRA3). A scanning electron microscope (SEM) image of the pristine ramie fabric and LCRF is shown in Figure 4a–c and Figure 4d–f, respectively. Figure 4a,d illustrate that the pristine fabric experiences slight shrinkage after carbonization, which would be attributed to the release of CO_2_, CO and H_2_O during the laser engraving. It can be seen from Figure 4e that some fibers seem to stick together, which is distinguished from the bast fibers carbonized in the oven [43] and may be related to the wrapping of CMC on ramie fibers.

Figure 5 showed the Raman spectrum and X-ray photoelectron spectroscopy (XPS) spectra of pristine ramie fabric and LCRF. Raman spectrum was measured with a Raman spectroscopy (WITec, Apyron, Beijing, China) equipped with a 532 nm laser wavelength. After laser engraving, the typical G-band at ~1334 cm^−1^, and D-band at ~1582 cm^−1^ could be seen clearly in Figure 5a, which could possibly indicate the graphite structure [43,44]. The XPS test (Thermo Fisher Scientific K-Alpha, Waltham, MA, USA) was conducted to analyze the element information in detail (Figure 5b). The peaks of C_1s_ and O_1s_ are found in the pristine ramie fabric while C_1s_, O_1s_ and Na_1s_ are found in the LCRF due to the addition of CMC. Further, the C/O atomic ratio of LCRF (2.81) is higher than that of the pristine ramie fabric (2.04), confirming the efficient loss of O atoms from the ramie fabric and thus the successful carbonization of the laser-engraved area [53], which is consistent with the conclusion of the Raman analysis. For further distinction, the analysis of the C_1s_ XPS spectra revealed four dominant peaks: sp^2^ C–C bond at ~284.5 eV, C–O bond at ~286.0 eV, C=O bond at ~288.5 eV, and COO bond at ~289.5 eV for the two samples [54], as illustrated in Figure 5c,d. It can be seen clearly that the content of the C–C bond increased from 18.57% to 69.43% and the content of the C–O bond decreased from 66.67% to 13.93%. The sp^2^ C–C peak became dominant in LCRF, and the carbonation degree was comparable to that in the laser-carbonized materials (e.g., wood [38,39], nanocellulose [52], lignin/PEO film [55]). As a result, the ramie fabric with CMC treatment can be carbonized to a high degree during laser engraving.

### 3.3. LCRF-Based Sensors

#### 3.3.1. Strain Sensor

In order to investigate the strain sensitivity, we applied two kinds of strains, compressing and tensioning, to the LCRF sensor and observed that the resistance of the strain sensor decreased or increased accordingly. This phenomenon can be explained by the compression and tension of carbonized fiber with cracks in LCRF (Figure 6a) [46,47]. By bending the inner surface, the sensor is under compressive stress, and the crack gap in the fibers narrows, which results in decreasing the resistance of the strain sensor. On the contrary, by bending the outer surface, the sensor was under tensile stress, and the crack gap in the fibers widens, which resulted in increasing the resistance of the strain sensor. The following experimental phenomena of the pre-bending test can be explained by this mechanism of cracks on the fibers. Figure 6b shows the relationships between the number of pre-bending cycles (radius of curvature was about 2.15 cm, representing 1.5% strain) and the normalized resistance change. Due to the cracks generated in the fibers, the resistance of the sensor in the natural state increased at first. After about 50 pre-bending cycles, the resistance of the sensor in its natural state increased by more than 31% and remained stable, so the LCRF-based strain sensors were all trained for 50 cycles.

As illustrated in Figure 7a,b, the values of GF for compression or tension are 128.9 or 136.3, respectively, higher than the reported bast fabric strain sensors carbonized in the oven [42,44]. Figure 7c exhibits that the sensor generates regular resistance change for bending strain and the response time was approximately 0.22 s or 0.27 s for tension or compression, respectively, which can meet the general sensing requirements of many daily motions. To evaluate the mechanical robustness and the reliability of the strain sensors, the resistance of the sensor was recorded during 5000 bending–unbending cycles (radius of curvature was about 2.93 cm, representing 1.1% strain, Figure 7d). After bending, the change of resistance still maintained stability, suggesting that the sensor is reliable for strain detections.

We further investigated the application potential of the flexible strain sensor in the field of human-motion detection. As fingers are able to perform many complex human actions, we attached the strain sensor to an index finger for the potential application in wearable electronics (Figure 7e). When the finger bent and stretched rapidly, the sensor could respond quickly and produce a short pulse signal, indicating good performance in rapid monitoring of human motion. Moreover, we laid another strain sensor on the folding line of a book as well in order to investigate it as a folding-type sensing device (Figure 7f). By measuring the resistance at the different angles θ, the electromechanical properties of sensors were investigated. We observed the resistance variation of the folding-type sensor by opening the book at 15°, 45°, 90°, 135°, and 180° step by step (each step was held for about five seconds) and then closing gradually from 180° to 135°, 90°, 45°, and 15°. When we opened the book from 15° to 180°, the resistance of the sensor increased from ~15.3 kΩ to ~20.3 kΩ. Then, the book was closed from 180° to 15°, while the resistance of the sensor decreased from ~20.3 kΩ to ~15.3 kΩ. At each turning angle, the folding-type sensor could respond rapidly and the resistance of the sensor could be maintained under the same angle. Similarly, we can also use this folding-type sensor to measure the changes in mechanical arm angle, suggesting an attractive perspective application in robots.

#### 3.3.2. Humidity Sensor

Humidity sensors are widely used in industrial instruments and automation, agriculture, medical care, and climate monitoring [56,57,58]. Taking advantage of the good water absorption of ramie fabric, the humidity sensor is composed of a laser-carbonized interdigital electrode and uncarbonized ramie fabric, as shown in Figure 8a. The capacitance value variations of the humidity sensor are recorded and depicted when the RH was increased from 11% to 97%. When RH = 11%, the capacitance value was 12.2 pF, and when RH = 97%, the capacitance value rose to 303.8 pF. The values of S for RH ranging from 0–57% or 57–100% are 5.17 or 55.26, respectively, higher than some reported natural fabric (cotton and wool fabric) humidity sensors (0.34–20.12) [59,60,61]. The interdigital electrode on ramie fabric had good flexibility and in order to evaluate the mechanical robustness and the reliability of the electrode, the capacitance value of the electrode was recorded during 5000 bending–unbending cycles (radius of curvature was about 1.52 cm, Figure 8b). The result showed that the capacitance value of the electrode only decreased by less than 4.4%, suggesting that the electrode had good reliability. Moreover, when the humidity is alternately converted from 45%RH (in the air) to 84%RH, the sensor generates a regular capacitance signal, as shown in Figure 8c. When the relative humidity in the test container is 84%RH, the capacitance needs about 900 s to reach equilibrium, and it takes about 1000 s to recover in the air. As a result, the humidity sensor’s response/recovery time between 45% and 84% RH is about 900 s/1000 s.

For real-time humidity detections, human breathing and blowing were monitored. As shown in Figure 8d, the humidity sensor was respectively close to the nose and mouth of the human body to monitor the water vapor generated by the human body in real-time. When monitoring human breathing, the humidity sensor can distinguish between normal breathing (about 12 times per minute) and deep breathing (about six times per minute) according to the change of capacitance. Furthermore, when blowing air (about five times per minute) onto the humidity sensor, the capacitance value will increase in about 3 s and recover in about 10 s. These results suggest the potential applicability for human-health monitoring. In order to use the patterning advantages of this laser-engraving method, a 3 × 3 humidity sensor array was fabricated to detect moving wet objects (Figure 8e). When a bared finger (3 cm from the array) is passed over the sensor array at different speeds, the capacitance value of each sensor on the array changes differently. When the finger speed was 1 cm/s, more water vapor entered the sensor, resulting in a greater capacitance increase and when the finger speed upgraded to 5 cm/s, less water vapor entered the sensor, resulting in less capacitance increase. We also used a finger with a rubber glove to pass over the array and the capacitance value of the humidity sensor did not change. As a result, these humidity sensors can provide the potential for human-health monitoring and speed detection of wet objects in real-time.

## 4. Conclusions

In summary, we successfully developed a simplified, cost-effective and environmentally friendly method for engraving ramie fabric directly by laser under an ambient atmosphere to prepare strain and humidity sensors. The addition of CMC to ramie fabric could effectively avoid ablation during laser engraving and the gained LCRF had good conductivity (65 Ω sq^−1^). The laser-engraving process allows for patterning on the fabric, and thus the fabrication of strain and humidity sensors. The fabric strain sensors feature a high GF of ≈128.9 (compressive strain), and ≈136.3 (tensile strain) and remarkable reliability and durability (>5000 cycles). The fabric humidity sensors have good flexibility and can detect the moving speed of wet objects in real-time. This low-cost laser-engraving carbonizing method can be used in fabricating electronic components on natural bast materials on a large scale.

## Figures and Tables

**Figure 1 micromachines-13-01309-f001:**
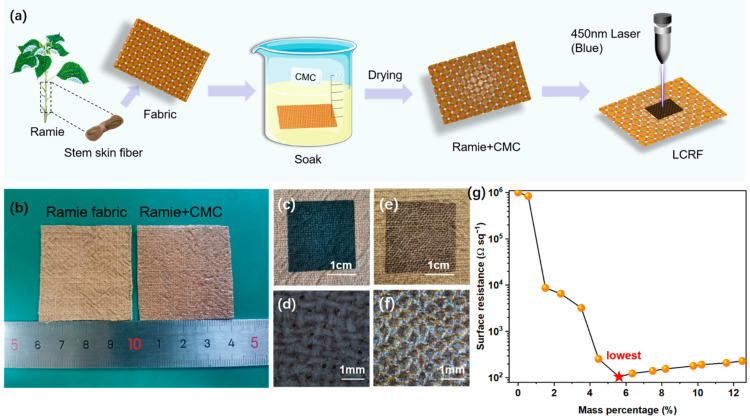
(**a**) Schematic illustration of the fabrication for LCRF. (**b**) photographs of untreated ramie fabric and ramie fabric treated with CMC. (**c**,**d**) surface morphology of carbonized ramie fabric treated with CMC. (**e**,**f**) surface morphology of carbonized untreated ramie fabric. (**g**) surface resistance of LCRF adding CMC with different mass percentages.

**Figure 2 micromachines-13-01309-f002:**
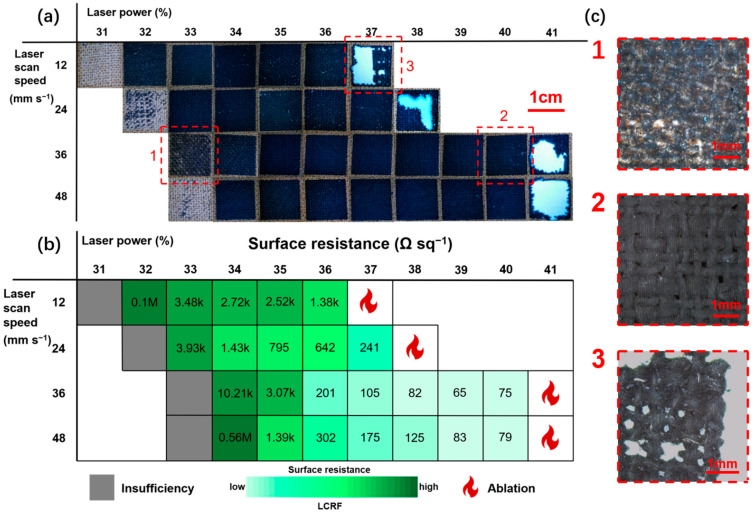
(**a**) LCRF engraved by different laser scan speed and power and (**b**) their surface resistance. (**c**) different LCRF for three different laser scan speeds and power.

**Figure 3 micromachines-13-01309-f003:**
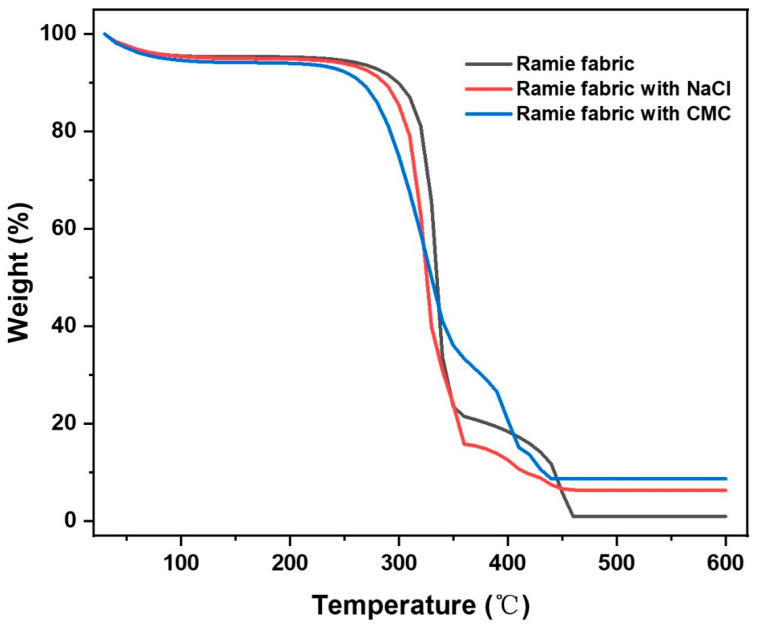
Thermogravimetric curves of different ramie fabrics in air.

**Figure 4 micromachines-13-01309-f004:**
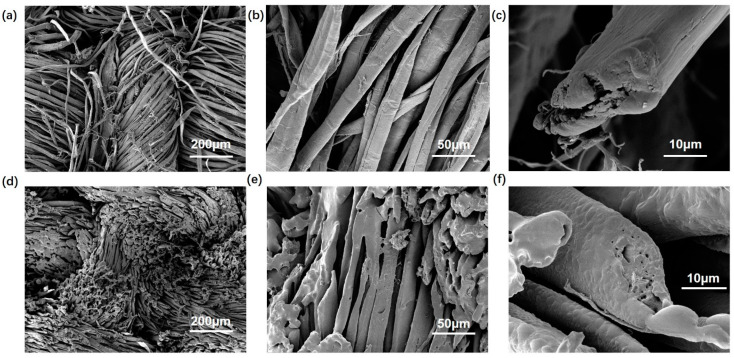
(**a**–**c**) SEM images of carbonized pristine ramie fabric. (**d**–**f**) SEM images of LCRF.

**Figure 5 micromachines-13-01309-f005:**
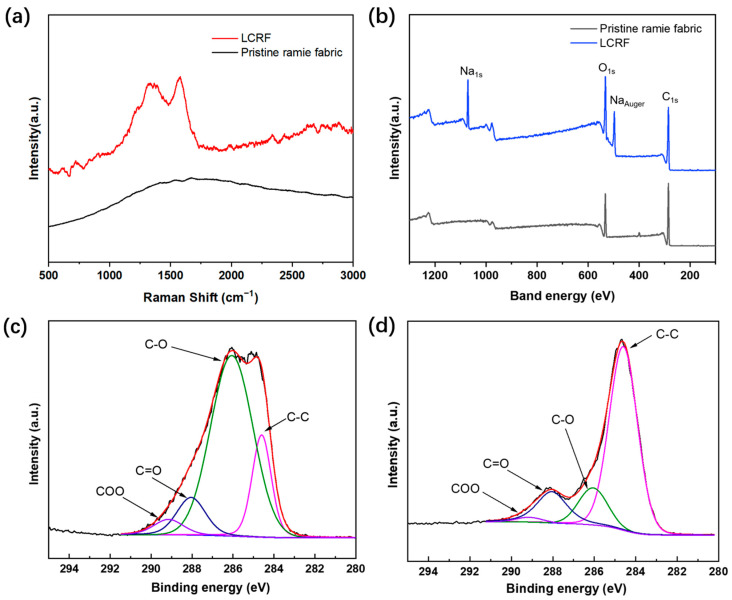
(**a**) Raman spectrum of pristine ramie fabric and LCRF. (**b**) XPS spectra of pristine ramie fabric and LCRF. (**c**) The detailed element information of the C_1s_ XPS spectra for pristine ramie fabric. (**d**) The detailed element information of the C_1s_ XPS spectra for LCRF.

**Figure 6 micromachines-13-01309-f006:**
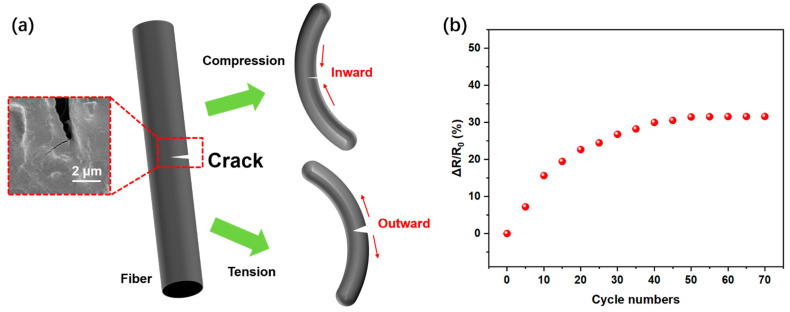
(**a**) Sensing mechanism of LCRF-based strain sensor. (**b**) Normalized resistance changes of sensor in the 70 pre-bending cycles.

**Figure 7 micromachines-13-01309-f007:**
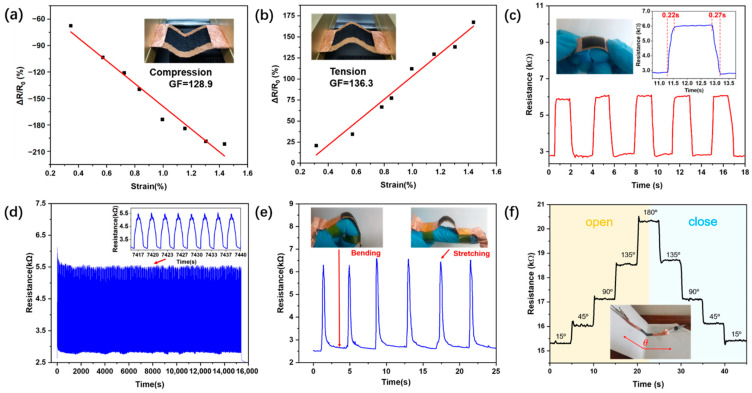
Performance of the sensor for detecting bending strains. Relative resistance changes as a function of applied strain. (**a**,**b**) are in accordance with the different bending directions (inward deflection (compression) or outward deflection (tension), respectively). (**c**) Response time of the sensor under bending strain. (**d**) Resistance change of the strain sensor during application of 5000 bending–unbending cycles. (**e**) Relative resistance changes during the finger bending tests. (**f**) Resistance change of the sensor under opening (orange area) and closing (blue area) the book states.

**Figure 8 micromachines-13-01309-f008:**
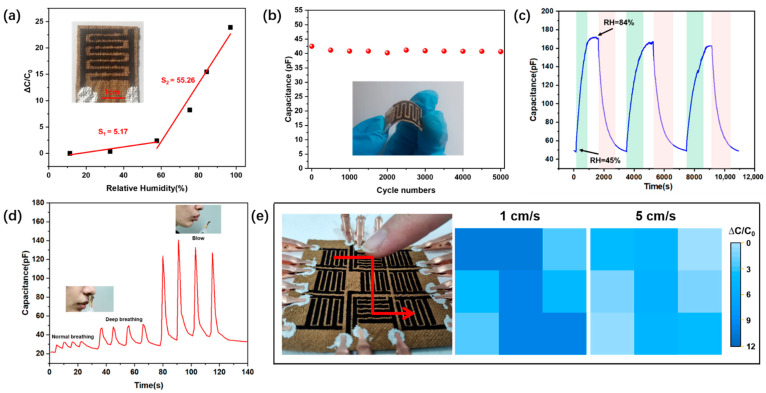
Performance of the sensor for detecting humidity. (**a**) Laser engraved interdigital electrode pattern and its relative capacitance changes as a function of relative humidity. (**b**) Capacitance change of the electrode during application of 5000 bending–unbending cycles. (**c**) Capacitance change of the humidity sensor under alternative humidity change from 45% to 84%. (**d**) Capacitance change of humidity sensor under human blowing and breathing. (**e**) Speed detection of a moving finger using a 3 × 3 humidity sensor array.

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
