# Peer review of "Ramie Fabric Treated with Carboxymethylcellulose and Laser Engraved for Strain and Humidity Sensing"

_micromachines, 2022, doi:10.3390/mi13081309_

Round 1

Reviewer 1 Report

The manuscript titled “Ramie Fabric Treated by Carboxymethylcellulose and Laser 3 Engraved for Strain and Humidity Sensing” discusses the use of laser engraving on a natural fiber based Ramie fabric for creating wearable sensors and the applications for human motion detection. The paper is organized well, with good results. However, there are plenty of grammatical errors and spelling mistakes. It is recommended that this paper be published after the following queries have been addressed. 

1.       Multiple spelling errors such as glue in line 91, page 2 of 12, stain on line 296

2.       For figure 2, what are the dimensions of the samples that are shown in each picture? Please add scale bars in figure 2c.

3.       Since this paper specifically focuses on the human motion part, please cite additional techniques that researchers have used to create these types of fabric/textile based wearable sensors and comment/compare at a high level what are the advantages and disadvantages of each process used. For example below are some papers that I have downloaded recently you could potentially some or all of these papers or find a review paper:

a.       Fu, Qingjin, et al. "Emerging cellulose-derived materials: a promising platform for the design of flexible wearable sensors toward health and environment monitoring." Materials Chemistry Frontiers 5.5 (2021): 2051-2091. (pyrolysis)

b.       Li, Yuanqing, et al. "Highly flexible strain sensor from tissue paper for wearable electronics." ACS sustainable chemistry & engineering 4.8 (2016): 4288-4295. (pyrolysis)

c.       Li, Yuan-Qing, et al. "Multifunctional wearable device based on flexible and conductive carbon sponge/polydimethylsiloxane composite." ACS applied materials & interfaces 8.48 (2016): 33189-33196.

d.       "Ultrahigh sensitivity wearable sensors enabled by electrophoretic deposition of carbon nanostructured composites onto everyday fabrics." Journal of Materials Chemistry C 10.5 (2022): 1617-1624. (electrophoretic deposition)

e.       "Multifunctional conductive hydrogel-based flexible wearable sensors." TrAC Trends in Analytical Chemistry 134 (2021): 116130. (hydrogel)

4.       What is the ‘spot size’ or the width of the sample engraved in each pass of the laser? Is there any overlapping in the areas in consecutive passes? If the width is 10 units for example, is the surface resistivity uniform at the center line as well as closer to the edges?

5.       Page 6 of 13, lines 201 to 223: How many samples were tested in each case? While average values of resistance is mentioned (4128 for Nacl and 65 for

6.       What is microstructure of the ramie fabric? Section 3.3.1 What type of a weave is it and how would the weave affect the sensing performance.

7.       What is the sensing mechanism – looking at the pictures, it seems like it is plain weave or something like a 2x1/4x1 weave. Is the resistance change only due to piezoresistivity at the nanoscale or also at the microscale?

8.       The resistance numbers mentioned in lines 300-305 do not match with the graph in figure 6(f). 14.7 kOhms appears to be wrong.

9.       Figure 6(e) is not very clear…..Show the state of the finger at baseline resistance and then at increased resistance. For the stretching case why is the sensor in a semi-circular loop?

Reviewer 2 Report

This manuscript reports a ramie fabric based strain and humidity sensor which is treated by the laser technical and carboxymethylcellulose. It has potential applications in multifunctional flexible electronics and wearable e-textile. However, some key issues need to solve.

1, The logic of writing may be a little confusing and the manuscript structure may need to be reorganized. For example, data for NaCl treated fabric as a reference sample have been presented and described in the TG figure, but the steps of NaCl treatment are described behind.

2, The author claims that “Fig. 4a, d illustrate that the pristine fabric experiences a surface shrinkage after carbonization, and the fibers became thinner shown in Figure 4b, e and Figure 4c, f,…” There is no obvious diameter thinning from the SEM images. It is suggest that the authors could draw the above conclusions by measuring the fineness of fibers before and after the treatment.

3, The authors’ used laser to engrave interdigital pattern on the ramie fabric and obtained the LCRF. And then, silver glue was applied to bottom of the interdigital patterned LCRF, and gained the humidity sensor. In fact, according to this structural design, both the fabric and silver glue could impact the signal of the capacitive sensor. The author may need to use some additional experiments to distinguish which dielectric layer (fabric or silver glue) has a greater impact on the humidity sensor?

4, The response and recovery time of the sensor in Fig. 7c takes 900 s and 1000 s, but there are only few seconds when testing breathing. The author may need to explain in the manuscript why there is such a big difference.

5, Textile-based electronic attracts more attention recently, using fiber material to create humidity sensor has been demonstrated by others, which have close relevance to this work. The author may cite more corresponding references.

(1) Adv. Funct. Mater., 2019, 29(43): 1904549. https://doi.org/10.1002/adfm.201904549

(2) Sens. Actuators, B 2016, 230, 528. http://dx.doi.org/10.1016/j.snb.2016.02.108

(3) Sens. Actuators, B 2017, 252,697 http://dx.doi.org/10.1016/j.snb.2017.06.062

(4) Text. Res. J. 2021, 91 (3-4), 398-405. https://doi.org/10.1177/0040517520944495

Reviewer 3 Report

This paper describes a simplified, cost-effective and environment-friendly method for engraving ramie fabric (a kind of bast fabric) directly by laser under ambient atmosphere. Further, the prepared fabrics are designed into strain and humidity sensors. The following concerns should be addressed before publication.

1.      Authors demonstrated CMC improvs the conductivity of LCRF significantly. Why don’t authors directly use cellulose fabrics for laser engraving treatment, instead of ramie fabrics treated with CMC?

2.      What is the response time for the capacitance-type humidity sensor?

3.      Authors demonstrated the humidity sensor for human respiration detection. In normal breathing, normal respiration rates for an adult person at rest range from 12 to 16 breaths per minute. However, in Figure 7d, it shows only three times within 1 minute, this is not a “normal breath” state. Is it because the response speed is not fast enough to measure the real-time respiration signal? Please clarify.

4.      In Fig. 6e, does the ascending part in the curve means bending, and descending part means recovery? It is suggested to mark the arrows and labels more clearly.

Round 2

Reviewer 3 Report

I still have the following concerns about the revised manuscript. 

1.      What is the response time for the capacitance-type humidity sensor?

2.      Authors demonstrated the humidity sensor for human respiration detection. In normal breathing, normal respiration rates for an adult person at rest range from 12 to 16 breaths per minute. However, in Figure 7d, it shows only three times within 1 minute, this is not a “normal breath” state. Is it because the response speed is not fast enough to measure the real-time respiration signal? Please clarify.

Author Response

Comment 1: What is the response time for the capacitance-type humidity sensor?

Response 1: 

Thank you for your comment.

1. The response and recovery times are defined as the time required to reach 90% of the change of sensor capacitance. During the test, humidityis alternately converted from 45%RH (in air) to 84%RH (in environment of saturated KCl solution). Humidity sensor’s response/recovery time between 45% and 84% RH is about 900 s/1000 s.

2. We have added the definition of response/recovery time in the manuscript:

“The response and recovery times are defined as the time required to reach 90% of the change of sensor capacitance (in line 197-198).”

3. Description about response time of the humidity sensor can been seen in Section 3.3.2:

“Moreover, when the humidity is alternately converted from 45%RH (in air) to 84%RH, the sensor generates regular capacitance signal, as shown in Fig. 8c. When the relative humidity in the test container is 84%RH, the capacitance needs about 900 seconds to reach equilibrium, and it takes about 1000 seconds to recover in the air. As a result, humidity sensor’s response/recovery time between 45% and 84% RH is about 900 s/1000 s (in line 346-351).”

Comment 2: Authors demonstrated the humidity sensor for human respiration detection. In normal breathing, normal respiration rates for an adult person at rest range from 12 to 16 breaths per minute. However, in Figure 7d, it shows only three times within 1 minute, this is not a “normal breath” state. Is it because the response speed is not fast enough to measure the real-time respiration signal? Please clarify.

Response 2:

Thank you for your comment.

1. According to your suggestion,we have retested the human breathing and blowing. In this test, the frequency of normal breathing was about 12 times per minute. Because of the slow recovery speed of the sensor, the capacitance of the sensor cannot return to the initial value during continuous breathing. As a result, the relative change of capacitance value during continuous breathing is small (ΔC/C0 = 28%), but the sensor can still measure the real-time respiration signal.

2. We added the description of the frequency for breathing and blowing in the manuscript:

“When monitoring human breathing, the humidity sensor can distinguish between normal breathing (about 12 times per minute) and deep breathing (about 6 times per minute) according to the change of capacitance. Furthermore, when blowing air (about 5 times per minute) to the humidity sensor, the capacitance value will increase in about 3 seconds and recover in about 10 seconds (in line 362-366).”
